# Magnetic Characterization and Moderate Cytotoxicity of Magnetic Mesoporous Silica Nanocomposite for Drug Delivery of Naproxen

**DOI:** 10.3390/nano11040901

**Published:** 2021-04-01

**Authors:** Adriana Zeleňáková, Jaroslava Szűcsová, Ľuboš Nagy, Vladimír Girman, Vladimír Zeleňák, Veronika Huntošová

**Affiliations:** 1Institute of Physics, Pavol Jozef Šafárik University in Košice, Park Angelinum 9, 040 01 Košice, Slovakia; jaroslava.szucsova@student.upjs.sk (J.S.); lubos.nagy@student.upjs.sk (Ľ.N.); vladimir.girman@upjs.sk (V.G.); 2Institute of Chemistry, Pavol Jozef Šafárik University in Košice, Moyzesova 11, 040 01 Košice, Slovakia; vladimir.zelenak@upjs.sk; 3Center for Interdisciplinary Biosciences, Pavol Jozef Šafárik University in Košice, Jesenna 5, 040 01 Košice, Slovakia; veronika.huntosova@upjs.sk

**Keywords:** iron oxide magnetic nanoparticles, cytotoxicity study, drug delivery

## Abstract

In this study, we describe the magnetic and structural properties and cytotoxicity of drug delivery composite (DDC) consisting of hexagonally ordered mesoporous silica, iron oxide magnetic nanoparticles (Fe_2_O_3_), and the drug naproxen (Napro). The nonsteroidal anti-inflammatory drug (NSAID) naproxen was adsorbed into the pores of MCM-41 silica after the ultra-small superparamagnetic iron oxide nanoparticles (USPIONs) encapsulation. Our results confirm the suppression of the Brownian relaxation process caused by a “gripping effect” since the rotation of the whole particle encapsulated in the porous system of mesoporous silica was disabled. This behavior was observed for the first time, to the best of our knowledge. Therefore, the dominant relaxation mechanism in powder and liquid form is the Néel process when the rotation of the nanoparticle’s magnetic moment is responsible for the relaxation. The in vitro cytotoxicity tests were performed using human glioma U87 MG cells, and the moderate manifestation of cell death, although at high concentrations of studied systems, was observed with fluorescent labeling by AnnexinV/FITC. All our results indicate that the as-prepared MCM-41/Napro/Fe_2_O_3_ composite has a potential application as a drug nanocarrier for magnetic-targeted drug delivery.

## 1. Introduction

Porous materials, their nature, and properties are often related to the inclusion phenomena. The inclusion chemistry of porous materials represents their extraordinary feature for which the porous materials are widely used in different applications, such as adsorption, separation, ion exchange, drug delivery, etc. [1,2,3]. The inclusion chemistry is being intensively investigated using different types of porous materials, including zeolites, porous polymers, porous carbons, oxides, coordination polymers [3]. Among different porous materials, mesoporous silica has been extensively studied in recent years due to its interesting properties and structural features. Large surface areas and pore volumes of mesoporous silicas, together with the narrow and controlled pore-size distribution and their rich intrapore chemistry, make these materials very attractive as supports for different applications [4,5,6,7].

Extraordinary properties of ordered mesoporous silica materials have stimulated research in areas that include fundamental studies on sorption and phase transitions in confined spaces, ion exchange, formation of various metal clusters, and loading different guest molecules, including biomolecules. Among different possible applications, one of the most promising is the use of mesoporous silica materials in biomedical applications and particularly as drug delivery systems (DDSs) [8,9,10,11,12]. Mesoporous silica DDSs are especially useful for the incorporation of drug molecules with a lack of specificity and solubility. It is well known that more than 40% of new drugs identified through combinatorial screening are poorly water soluble [13]. These drugs are characterized by low adsorption and poor bioavailability, and they may lead patients to take high doses of the drug to achieve sufficient therapeutic effects. Drug delivery of hydrophobic drugs using DDSs based on mesoporous silica nanoparticles can overcome many drawbacks [14]. The first mesoporous silica-based DDS was MCM-41 material [15]. The release study of ibuprofen loaded into mesopores of MCM-41 showed sustained drug release over three days. This very first study of the silica as a DDS showed that delivery is based on dissolution/diffusion of the confined drug. The release of drugs from the silica mesopores is a complex process that depends on many parameters such as the different solubility of guest molecules in the loading solvent, the different diffusion rates throughout the pores, and/or the strength of the interaction between loaded molecules and silica. However, at present, more sophisticated systems can be prepared that hamper an undesired (premature) release (leakage) of loaded compounds and the release can be triggered on demand, using various physical or chemical stimuli [16,17,18,19,20,21,22]. Thus, with the use of drug delivery systems, the rate at which a drug is released and the location in the body where it is released can be controlled. Some systems can control both these parameters. Moreover, another positive feature of the mesoporous silica is that this DDS can accumulate within the tumor mass and transport anticancer drugs inside the cells.

In our previous works, we have prepared and described the different mesoporous silica particles that release drugs either on diffusion principle [23] or using external physical stimulus, e.g., light-driven drug release [24], redox driven drug release [25], or pH-driven drug release [26]. In addition to these stimuli, the magnetic field is another option to be used in vectored drug delivery and/or in delivery with low premature leakage. To date, many studies have demonstrated the preparation of DDSs containing magnetic nanoparticles for drug delivery or hyperthermia therapy [27,28,29,30,31,32,33,34]. Sousa et al. prepared Fe_3_O_4_/SBA-15 mesoporous nanocomposites by loading an iron precursor into the SBA-15 framework. The hyperthermia tests indicated that the Fe_3_O_4_/SBA-15 nanocomposites had the potential for localized hyperthermia treatment of cancer [35]. Other authors described the preparation of core/shell structured Fe_3_O_4_/SiO_2_ mesoporous nanoparticles for sustained drug release [36,37]. Giri et al. demonstrated that mesoporous silica nanorods capped with superparamagnetic iron oxide nanoparticles can be used as a stimuli-responsive and controlled-release delivery carrier [38]. Guest molecules such as fluorescein were encapsulated and released from the magnetic DDS [39]. Moreover, pH-sensitive magnetite mesoporous silica nanocomposites for controlled drug delivery and hyperthermia were recently reported [40]. The confined magnetic nanoparticles inside mesoporous silica often show superparamagnetic behavior [41,42]. Shevtsov investigated mesoporous silica MCM-41 nanoparticles impregnated with iron as a nanocarrier system for drug delivery into tumor cells. Magnetization study confirmed the superparamagnetic behavior at room temperature and the saturation magnetization M_s_ with a value of 7.4 emu/g. Bio-distribution studies with magnetization measurements demonstrated increased and dose-dependent retention of MSNs in tumor tissues [42]. In the light of the above-mentioned studies and advantages of magnetic mesoporous silica, in the present study, we focused on the investigation of drug delivery composite (DDC) system of mesoporous silica containing the iron oxide nanoparticles and naproxen drug. The detailed magnetic behavior of this system in the temperature range of 2–300 K in external magnetic field up to 5 T was described and cytotoxicity of the studied DDC consisting of MCM-41 silica, magnetic nanoparticles Fe_2_O_3,_ and naproxen (sample MCM-41/Napro/Fe_2_O_3_) was evaluated.

## 2. Materials and Methods

Tetraethoxysilane (hereafter denoted as TEOS, Cat. No: 86578, Sigma-Aldrich, St. Louis, MO, USA) was used as the silica source, cetyltrimethylammonium bromide (CTAB, Cat. No: H5882, Sigma-Aldrich, St. Louis, MO, USA) was used as the structure-directing agent. Iron nitrate nonahydrate, Fe(NO_3_)_3_∙9H_2_O, (Cat. No: 254223, Sigma-Aldrich, St. Louis, MO, USA), was used for the preparation of hematite superparamagnetic iron oxide nanoparticles (SPIONs). All reagents, including naproxen, were obtained from Aldrich and used as received without further purification.

### 2.1. Material Preparation

MCM-41 mesoporous silica matrix was synthesized according to Qiang et al. [43] in the molar ratio of TEOS/CTAB/NaOH/H_2_O equal 1/0.12/0.33/601.3. The hematite-containing nanocomposite material was prepared by the wet impregnation of mesoporous matrix MCM-41 by 1 M aqueous solution of Fe(NO_3_)_3_.9H_2_O. The obtained solid product was gently washed with water and air dried. The dry product was thermally treated at 773 K for 6 h, leading to the decomposition of the nitrate to the iron oxide. The X-ray diffraction (XRD) analysis confirmed the hematite (α-Fe_2_O_3_) phase.

To load the mesoporous support with naproxen (Napro), 250 mg of the parent- or hematite-modified mesoporous silica, MCM-41 was suspended in the 40 mL of ethanolic solution of naproxen (10 mg/mL). Before the drug loading, the porous matrix was dried at 150 °C for 3 h. The suspended matrices were stirred in the ethanolic solution of naproxen for 24 h at laboratory temperature preventing evaporation. Then, the solvent was evaporated till matrices remained wet and solvent soaked into the pores, to maximize the mass of loaded naproxen. The obtained products were filtered off three times and then washed with ethanol and dried at laboratory temperature. The prepared samples were denoted as sample 1—blank MCM-41, sample 2—MCM-41/Napro, sample 3—MCM-41/Fe_2_O_3_, and sample 4—MCM-41/Napro/Fe_2_O_3_. The amount of the loaded naproxen was determined thermogravimetrically and represented 302 mg of the drug in 1 g of sample 2—MCM-41/Napro and 276 mg of the drug in 1 g of sample 4—MCM-41/Napro/Fe_2_O_3_.

### 2.2. Drug Release

The release experiment was realized in physiological intravenous saline solution (0.9% NaCl). During the experiment, 100 mg of sample 2 (MCM-41/Napro) or sample 4 (MCM-41/Napro/Fe_2_O_3_) was suspended in 10 mL of physiological solution. The mixtures were subsequently stirred at 37 °C for 72 h. The concentration change in buffer medium was monitored in selected time intervals of 1 h, 2 h, 5 h, 7 h, 9 h, 24 h, 48 h, and 72 h by HPLC (High Performance Liquid Chromatography).

### 2.3. Structural and Magnetic Characterization

The transmission electron microscopy (TEM) micrographs were taken with a JEOL 2100 microscope (Tokyo, Japan). A copper grid coated with holey carbon support film was used to prepare samples for the TEM observation. The TEM images obtained were processed in Image J (version 2.1.4.7, National Institutes of Health, Bethesda, MD, USA) to obtain a Fourier diffractogram through fast Fourier transform (FFT) analysis. This FFT analysis essentially transforms the greyscale distribution function of a TEM image into a frequency distribution function, from which a diffraction pattern is obtained that can be used to determine the periodicity and symmetry of the pore structure.

Diffraction measurements were carried out by synchrotron radiation of energy 60 and wavelength λ = 0.123980 Å at PETRA III accelerator at DESY, Hamburg, Germany. Kapton capillaries were filled with powder samples. The obtained diffraction patterns were processed via FIT2D software, employing CeO_2_ as a calibration standard. Particles’ size was determined by the Scherer formula.

The particle-size distribution was measured by dynamic light scattering (DLS) using ZetaSizer Nano S, Malvern Instruments (Malvern, UK). The powder sample was dispersed in deionized water and homogenized by ultrasonic breaker for 5 min. 

The porosity and specific surface area of materials were determined at 77 K by nitrogen adsorption/desorption using Quantachrome NOVA (Boynton Beach, FL, USA) 1200e surface area and a pore size analyzer. Prior to the experiments, the samples were outgassed at 403 K for 24 h. The specific surface area (S_BET_) was estimated using the Brunauer–Emmett–Teller (BET) equation in a pressure range of 0.05–0.30, the pore-size distribution was calculated using the DFT (Density functional theory).

Infrared spectra were measured by Avatar FTIR spectrophotometer (GMI, Ramsey, MN, USA) in the range 4000–400 cm^−1^. Samples in solid state were analyzed in form of KBr pellets with the sample/KBr mass ratio of 1/100. Before pellets preparation, the KBr was dried in an oven at 600 °C for 3 h.

The magnetic measurements were performed on a commercial SQUID-based magnetometer (Quantum Design MPMS 5 XL, San Diego, CA, USA) over a wide range of temperatures (2–300 K) and applied DC fields (up to 50 kOe). The magnetic properties of the powder sample were changed by dispersing the powder in deionized water. Powder and liquid samples were encapsulated into plastic capsules and placed into plastic sample holders. 

### 2.4. In Vitro Cytotoxicity

#### 2.4.1. Cell Culture

The U87 MG (human glioma) cells were purchased from Cells Lines Services, Eppelheim, Germany. Cells were grown in culture medium (Dulbecco’s modified Eagle medium (D-MEM)) supplemented with 10% fetal bovine serum (FBS), L-glutamine (862 mg/L), sodium pyruvate (110 mg/L), glucose (4500 mg/L), and penicillin/streptomycin (1% *w*/*w*) in the dark at 37 °C, 5% CO_2_, and humidified atmosphere until 80% confluence (solutions were purchased from Gibco-Invitrogen, Life Technologies Ltd., Paisley, UK).

#### 2.4.2. MTT Assay

U87 MG cells were 24 h or 48 h incubated with 1, 10, 20, 50, and 100 µL/mL distilled water, sample 1, sample 2, sample 3, and sample 4 in complete cell culture media. Cell viability was assessed by MTT (3-(4,5-dimethylthiazol-2-yl)-2,5-diphenyltetrazolium bromide, Sigma-Aldrich, Darmstadt, Germany) assay detected at 560 nm and 750 nm by 96-well plate absorption reader (GloMax^®^-Multi+ Detection System with Instinct Software, Promega Corporation, Madison, WI, USA). A standard protocol for MTT assay was performed, i.e., 10 µL of 5 mg/mL MTT dissolved in phosphate saline buffer (PBS, pH = 7.4) were added into each well filled with 100 µL medium (alternatively, 100 µL of MTT were added into 1 mL media in Petri dish), and the plate was 1 h incubated in the dark at 37 °C; after 1 h, the cell culture medium was taken out from the well, and 200 µL (alternatively 1 mL in Petri dish) of dimethylsulfoxide (Sigma-Aldrich, Darmstadt, Germany) was added to each well to make dissolved formazan crystals.

#### 2.4.3. Fluorescence Microscopy

Cells were grown in glass coverslip bottom Petri dishes (35 mm, No. 0, MatTek, Ashland, MA, USA) at the density of 10^5^ cells. Substances at concentrations of 10 and 100 µL were administered 24 h before observation with an inverted LSM700 confocal microscope (Zeiss, Oberkochen, Germany) equipped with a 20× Fluar (NA = 0.75, ∞, Zeiss, Germany) and a CCD camera (AxioCam HRm, Zeiss, Oberkochen, Germany). Mitochondrial probe MitoTracker^®^ Orange CMTM/Ros (MTO, 0.2 µM, 15 min, ThermoFisher Scientific, Waltham, MA, USA) was excited by 555 nm CW solid-state laser, and the emission was detected >580 nm. Nuclei of cells were stained with 10 µg/mL Hoechst 33258 (Hoechst, 15 min, ThermoFisher Scientific, Waltham, MA, USA) that was excited with 405 nm laser, and its emission was detected in the range 410–490 nm. Phosphatidylserine was labeled with AnnexinV/FITC (5 µL from the detection Kit, 15 min, Annexin V-FITC Kit, Miltenyi Biotec, Bergisch Gladbach, Germany), which was excited at 488 nm and detected in the range of 500–540 nm. The fluorescence images were analyzed in Zen 2011 software (Zeiss, Oberkochen, Germany) or ImageJ software (National Institutes of Health, Bethesda, MD, USA). The bright-field images were detected with the same microscope.

## 3. Results and Discussion

### 3.1. Structural Characterization of Samples 1–4

Structural characterization of prepared samples by transmission electron microscopy (TEM) is presented in Figure 1, which shows the size and shape of the blank mesoporous silica MCM-41 (sample 1), sample MCM-41 with drug naproxen inside the pores (MCM-41/Napro; sample 2), sample MCM-41 with Fe_2_O_3_ nanoparticles inside the pores (MCM-41/Fe_2_O_3_; sample 3), and MCM-41 with Fe_2_O_3_ nanoparticles and drug naproxen inside the pores (MCM-41/Napro/Fe_2_O_3_; sample 4). Blank MCM-41 silica shows the regular porous system with hexagonal symmetry and the pore size around 3–4 nm (see Figure 1 (1A), sample 1). The mesoporous silica particles have a rod-like shape, with a length of about 350 nm and a width of about 150 nm (see Figure 1 (1B–4B, down)). After modification and naproxen loading the mesostructure, as evidenced by HRTEM (High-resolution transmission electron microscopy), remained unchanged. After the loading of Fe_2_O_3_ nanoparticles inside the porous system (sample 3), the regular hexagonal symmetry of MCM-41 silica is also retained (3A, 3B). Iron oxide magnetic nanoparticles embedded inside the MCM-41 matrix are visible as dark spots inside the silica grains in Figure 1 (3A and 4A).

From these observations, it is obvious that the silica matrix controls the growth of Fe_2_O_3_ MNPs (magnetic nanoparticles) and serves as a nanoreactor for the preparation of superparamagnetic iron oxide nanoparticles (SPIONs) with very low particle size distribution, determined by the size of mesochannels (3–4 nm).

The fast Fourier transform (FFT) spectra of the high-resolution images obtained on samples 1 and 2 (without and with drug in pores) are presented in Figure 2a,b. FFT spectra reveal an ordered set of reflections in frequency space with an overall sixfold symmetry. FFT converts the periodic pattern of real structure to corresponding symmetry of reflections in frequency space. Regarding our samples, the only structural feature repeated periodically is holes. Aware of this fact, the authors determined that the differences in FFT spots distances and shapes can be used for consideration about the distinction between both samples. In blank mesoporous silica of MCM-41, the presence of an ideal single set of six equidistant spots in the FFT suggests that the pores remain in a shape of perfectly ordered pattern without any directional symmetry deviation throughout the volume of the particle and are not internally divided into separate sub-domains of pores. On the other hand, in sample 2 with incorporated drug naproxen inside pores, the sharpness disparity of the spots distances around the center indicates that there is a little variance of the pore dimensions throughout the particle, particle, as confirmed by little difference in values of A, B, C (see Figure 2a,b). The loss of reflection’s sharpness also indicates that the change in pore size is not constant for a given direction but varies in a certain interval. This can be a result of both the partial alteration of mesopore walls during the treatment of the MCM-41 sample in the solution for 24 h during the drug impregnation and the drug loaded in mesopores. The FFT patterns provide useful information about structural changes, which can be correlated with drug loading into the pores of sample 2. To identify the iron oxide phase in the iron oxide modified MCM-41/Fe_2_O_3_ sample (sample 3), the wide-angle X-ray scattering was measured (Figure 2c). The presence of α-Fe_2_O_3_ (hematite; ICSD file No. 790007) was identified in the pattern of sample 3. It is worth noting that the diffraction peaks of the hematite phase are wide, due to the nanocrystalline character of the embedded SPIONs. Moreover, the diffraction peaks of hematite are overlapped by the signal coming from amorphous MCM-41 silica matrix background. Since sample 4 (MCM-41/Napro/Fe_2_O_3_) was produced from sample 3 by subsequent loading of drug naproxen, the presence of hematite magnetic nanoparticles in sample 4 is obvious. The phase composition of samples containing iron oxide magnetic nanoparticles prepared in a similar way in different mesoporous silicas was studied previously in our recent paper [44]. 

For the use of Nanoparticles in bio-inspired applications, nanoparticles with a lack of agglomeration are required. The tendency of the nanoparticles to aggregate in the liquid phase was studied using DLS (Dynamic light scattering) measurements (Figure 3a). As it can be inferred from Figure 3, the average particle size determined for sample MCM-41 by DLS was similar at pH = 2 and pH = 7, with the average value of 205 nm, which is in good agreement with the particle size calculated from TEM images. This outcome suggests that no agglomeration of MCM-41 particles took place in acidic or physiological pH. However, for samples 3 and 4 at pH = 2, the average size of the particles determined by the DSL method was larger than the average size determined for samples 1 and 2, which suggests that partial agglomeration of samples with magnetic nanoparticles at pH = 2 took place, which can be a result of the interparticle interactions. At low pH, the protonation of surface hydroxyls of MCM-41 silica and the presence of Cl^−^ anions (HCl was used to adjust the pH to 2) may lead to the non-covalent interactions between silica nanoparticles, which enhances the agglomeration.

Zeta potential (ZP) is an important parameter for the characterization of colloidal nanoparticle systems and provides information related to their stability under different conditions. Figure 3b displays the values of Zeta potential for samples 1–4 in pH = 2 and pH = 7. Results are very different for each pH value. The most stable system appears to be sample 3 (MCM-41/Fe_2_O_3_) at pH = 7, as indicated by the ZP value around −40 mV, which is an admissible characteristic value for stable colloidal systems. Sample 4 also exhibits a promising value of ZP, around −60 mV, i.e., in the range from −40 mV to −60 mV, the system is considered to have good stability. ZP values smaller than −30 mV indicates agglomeration and disintegration of the sample; these results can be observed in all samples exposed to an environment with pH = 2. The isoelectric points of hematite and silica are different (pH ~7.8 for hematite and pH ~2 for ideal bulk silica) [45,46,47] but may be a little shifted for different samples. Therefore, the zeta potential (ZP) is an interplay of the surface charges of both these systems. The silica matrix phase is in abundance. At pH = 7, silica particles are negatively charged while hematite positively. As a result, silica samples 1 and 2, containing no iron nanoparticles, have larger negative ZP than samples 3 and 4, containing hematite. The same holds for pH = 2. Since we are closer to the isoelectric point of silica (around pH = 2 for the bulk silica), the ZP significantly increases in comparison with pH = 7. Samples containing hematite are closer to zero ZP value characteristic for isoelectric point.

### 3.2. Textural Characterization of Samples 1–4

The properties of the samples and filling of the pores by hematite nanoparticles and/or naproxen molecules were well reflected by the textural characteristics of the samples, surface area, pore size distribution, and pore volume. Textural characteristics were obtained from nitrogen adsorption isotherms at 77 K. The adsorption/desorption isotherms are shown in Figure 4. The isotherm of pure MCM-41 silica can be characterized as of IVb type according to the IUPAC classification with a sharp step over a narrow range of relative pressures (P/P_0_ = 0.2–0.4) arising from the capillary condensation of nitrogen inside the mesopores. No adsorption/desorption hysteresis was observed. BET surface area of the MCM-41 sample was 820 m^2^/g with a total pore volume of 0.642 cm^3^/g. The pore-size distribution for the MCM-41 sample calculated using DFT theory was centered at 35 Å. The filling of the pores of the MCM-41 silica by hematite nanoparticles was reflected by the decrease of the total adsorbed volume of the nitrogen and a slight downshift of the capillary condensation step. However, the shape of the adsorption isotherm for the sample MCM-41/Fe_2_O_3_ did not change in comparison with the MCM-41 sample. The BET surface area for the sample MCM-41/Fe_2_O_3_ was 670 m^2^/g with a total pore volume of 0.516 cm^3^/g and the pore-size distribution calculated using DFT theory was centered at 32 Å. The loading of the naproxen molecules into MCM-41 and MCM-41/Fe_2_O_3_ led to the diminishment of the capillary condensation step, indicating that mesopores the samples are filled by the drug molecules. Only small adsorption at low relative pressures (below P/P_0_ = 0.2) was observed, due to adsorption in micropores. In accordance with these findings, the surface area, pore size, and pore volume decreased, and the respective textural parameters are summarized in Table 1.

### 3.3. Infrared Spectra 

The infrared spectra were used to confirm the naproxen loading by the porous MCM-41 and MCM-41/Fe_2_O_3_ samples. The infrared spectra are shown in Figure A1, Appendix A. For the pure silica sample MCM-41, the asymmetric ν_as_(Si-O-Si) and symmetric ν_s_(Si–O–Si) stretches were observed at 1066 and 794 cm^−1^, respectively [48,49]. The incorporation of hematite nanoparticles did not change the infrared spectrum and spectra of the samples MCM-41 and MCM-41/Fe_2_O_3_ are similar. It is worth noting that hematite nanoparticles are incorporated in the channels of the amorphous silica matrix, which dominates, and the stretching vibration of the hematite cannot be distinguished in the spectra. Moreover, the KBr technique used for measurements also diminishes this identification. The loading of the naproxen was reflected by several different absorption bands, such as the stretching vibration ν(C=O) of the carboxylic group of naproxen at about 1720 cm^−1^, the breathing vibrations of the aromatic rings in the range 1600–1500 cm^−1^ or γ(C–H) vibrations of naproxen in the region below 900 cm^−1^ [50]. The comparison of spectra of pure naproxen and spectra of samples 3 and 4 are shown in Figure A1c–e, Appendix A. The hematite NPs are hidden by the silica matrix. The IR beam is first absorbed by the silica walls on the surface. Therefore, the IR response of hematite nanoparticles embedded in silica is too weak to be observed. The absorption bands in the IR spectra in the region about 500 cm^−1^ are stronger for silica than hematite. Therefore, stretching vibration Fe-O of the hematite NPs cannot be distinguished in the spectra.

### 3.4. Magnetic Characterization of Samples 3 and 4

Samples 3 and 4 containing magnetic nanoparticles inside the pores were studied by SQUID magnetometry with the aim to analyze the potential of active targeting of drugs by the magnetic field. Figure 5 represents the temperature dependence of magnetization recorded in zero-field-cooling/field cooling (ZFC/FC) protocols at a low DC field of 100 Oe, which is typical for superparamagnetic systems. ZFC/FC behavior for powdered sample 3 (black squares) and sample 4 (red squares) is the same. On ZFC curves of both studied samples, the maximum temperature associated with blocking temperature T_B_ around 14 K was observed. It is clearly seen from Figure 5 that at a temperature above T_B_ (T > T_B_), the particle’s magnetic moments can freely fluctuate through the energy barrier of the magnetic system, leading to a superparamagnetic state. Conversely, below T_B_ (T < T_B_), the magnetic moments are blocked in the direction of the external magnetic field, which is documented by the high irreversibility of ZFC and FC curves and by the existence of hysteresis below T_B_. Identical behavior of powdered sample 3 (MCM-41/Fe_2_O_3_) and sample 4 (MCM-41/Napro/Fe_2_O_3_), along with the y-shifted ZFC/FC spectra due to different amount of magnetic nanoparticles per gram, confirmed that total magnetic properties in composites are determined only by the properties of Fe_2_O_3_ nanoparticles loaded within nanopores.

Magnetization curves of samples 3 and 4, measured at 2 K and 300 K (see Figure 6) confirmed the superparamagnetic behavior above blocking temperature (300 K) and blocking process of particles magnetic moments below T_B_ (2 K). The values of coercivity estimated from low temperatures hysteresis loops (measured at 2 K) were H_C_ = 2040 Oe and H_C_ = 3175 Oe for MCM-41/Fe_2_O_3_ and MCM-41/Napro/Fe_2_O_3_, respectively. The high value of magnetic moment m_p_ = 125 μ_B_ and m_p_ = 139 μ_B_ was estimated for both samples using the Langevin function [48] (red line on Figure 6). Hematite is the most stable and extensively studied iron oxide. Bulk hematite is a crystal with a corundum structure containing two sublattices creating an antiferromagnetic system. The magnetic moment of bulk hematite and microsized particles is very small and therefore unsuitable for biomedical applications. With decreasing size of hematite particles below 10 nm, the magnetic moment of particles drastically increases. They show slight canting considering the basal plane, resulting in a magnetic moment, which originates from the superexchange interaction. Additionally, was observed the absence of Morin temperature (T_M_) for the particles with size 8–20 nm. Due to the suppression of the Morin transition in small hematite nanoparticles (below ~20 nm), only a high-temperature phase can exist, in which the net magnetic moment originates in small spin canting away from antiferromagnetic alignment. Thus, with regard to the above, the magnetic moment of nanosized hematite is comparable to other forms of iron(III) oxide. 

The value of magnetization (magnetic moment per gram) can be observed as very small from Figure 6 for S3 and S4. However, it is necessary to note that the absolute value of measured magnetization is “distorted” because the measured amount counts the nanoparticles together with the silica. Thus, the actual magnetization value of the hematite particles calculated without silica is slightly higher. We calculated the value of magnetization per hematite particles with respect to the 15% amount of particles inside MCM-41 in sample 3. The calculated value of 8.6 emu/g at 300 K and 32.6 emu/g at 2 K is comparable with other magnetic systems [30,31,32].

To identify the Brownian and Néel contribution to the total relaxation mechanism (1/τ = 1/τ_B_ + 1/τ_N_), the simple comparison of magnetic properties of powdered and liquid diluted sample 4 was realized (MCM-41/Napro/Fe_2_O_3_ diluted in saline solution with pH = 7 and concentration of magnetic nanoparticles 10 mg/mL, Figure 7a,b). As is can be derived from Figure 7a, the ZFC/FC curves of the powdered and liquid samples are very similar. The absolute values of ZFC magnetization of powdered sample 4 (black squares, left y-axis) are almost three times higher than that diluted in saline solution sample 4 (magenta squares, right y-axis) due to the different amount of magnetic nanoparticles per gram. Moreover, the behavior of ZFC/FC curves is almost identical, suggesting that the contribution of the Brown relaxation process to total relaxation is very small. To identify the strength of inter-particle interactions, the AC susceptibility data were analyzed in the frame of Néel–Arrhenius law (Equation (1)), Vogel–Fulcher law (Equation (2)), and critical-slowing-down law (Equation (3)) [48].

The relaxation time depends on the different above-mentioned laws by Equations (1)–(3) [48] as follows:(1)$$\tau =\tau_{0}e^{\frac{E_{A}}{k_{B}T}},$$
where *τ*_0_ denotes the pre-relaxation constant, *E_A_* stands for the activation energy (*E_A_ = KV*), K is the effective magnetocrystalline anisotropy constant, V is the volume of the magnetic particles, and *k_B_* represents the Boltzmann constant and *T* is the temperature. With increasing interaction strength, the relaxation time does not follow the Néel–Arrhenius law (Equation (1)), and authors employed the Vogel–Fulcher law (Equation (2)) for the data evaluation as follows:(2)$$\tau =\tau_{0}e^{\frac{E*}{k_{B}(T-T_{0})}},$$
where *E** is the energy barrier (activation energy) modified by an effective contribution of the inter-particle interactions, and *T*_0_ is the parameter corresponding to the strength of the interactions. For even stronger interactions, the collective behavior of superspins predominated and so-called superspin glass state below the critical temperature for superspin ordering *T_SSG_* can be observed. In this case, the relaxation time is given by Equation (3) as follows:(3)$$\tau =\tau_{0}^{*}{\left(\frac{T-T_{SSG}}{T_{SSG}}\right)}^{-zv},$$
with the critical exponent *z**ν*, where ν is the critical exponent of the correlation length. 

The best results of our experimental data analyses were *τ*_0_ = 9.35 × 10^−14^ s and *T*_0_ = 2.28 K and *E_A_*/*k_B_* = 457 K provided by the data fit according to the Vogel–Fulcher law (Equation (2), see Figure 7d). The analysis of AC susceptibility and relaxation time dependence on the activation energy was conducted also on sample 4 diluted in saline solution with nanoparticle concentration of 10 mg/mL. Our results confirmed the suppression of the Brownian relaxation process caused by a “gripping effect” since the rotation of the whole particle encapsulated in the porous system of mesoporous silica was disabled. Therefore, the dominant relaxation mechanism in powder and sample diluted in saline solution is the Néel process, in which the nanoparticle’s magnetic moment rotation is responsible for relaxation.

### 3.5. In Vitro Cytotoxicity

#### 3.5.1. Biocompatibility of Sample 1–4 with U87 MG Cells

In vitro cytotoxicity was evaluated according to cell metabolic activity, which was measured with MTT assay. All studied substances did not significantly influence the viability of cells at studied conditions. A decrease in formazan production detected at 48 h after administration of 100 µL/mL concentrations was probably due to nutrition deficiency caused by a higher concentration of solvent (H_2_O). This decrease in metabolic activity of cells was not observed in samples 24 h after administration of these concentrations, which suggests that all studied substances were minimally cytotoxic to U87 MG cells (see Figure 8).

#### 3.5.2. Distribution of Samples 1–4 in U87 MG Cells

Bright-field images of U87 MG cells demonstrate morphology of U87 MG cells, which is not importantly changed in the presence of substances (samples 1–4, Figure 9). Nanoparticle aggregates (dark spots) can be observed in samples 3 and 4 at both concentrations. A higher number of dark spots was detected at 100 µL/mL concentrations. While at low concentrations (10 µL/mL), the nanoparticles form fewer clusters and are suggested to localize in high-contrasted perinuclear areas, the higher applied concentrations (100 µL/mL) of nanoparticles result in nanoparticle’s clusters formation (Figure 9). Created clusters are too big to pass the plasma membrane of the cells. The effect of size, shape, and surface charge on cellular uptake and subcellular distribution of nanoparticles was recently summarized in a review by Foroozandeh and Aziz [49]. In the present study, the nanoparticles’ clusters were adsorbed to the cell surface. However, it should be noted that naproxen transported by those nanoparticles can easily release and pass the plasma membrane of cells and small nanoparticles (<200 nm). Interestingly, round vesicular objects were detected in the cell’s cytoplasm at the presence of 100 µL/mL of sample 2. We hypothesize that these vesicular objects could be associated with autophagic vesicles and the process of cell detoxication. Autophagy was reported to be an important target in anticancer therapy, especially for the application of metallic nanoparticles that was reviewed in Cordani’s study [50]. With our detection limits, it is difficult to distinguish the presence of nanoparticles inside the vesicles. However, the formation of the vesicles due to naproxen interaction with cells should also be taken into consideration.

#### 3.5.3. Manifestation of Cell Death in U87 MG Cells in the Presence of Samples 2–4

A high level of nanoparticle biocompatibility with glioma cells was demonstrated above by means of cell morphology and metabolic activity. However, vesicle formation in cells suggests a certain degree of intoxication caused by the presence of the nanoparticles. MTT assay is closely connected with mitochondria in which formazan is produced [51]. For this reason, mitochondria of U87 MG cells were labeled with mitochondrial probe MTO. MTO is partially sensitive to mitochondrial membrane potential. Dissipation of mitochondrial membrane potentially induced by toxins can be observed in cells by decreasing MTO fluorescence intensity until the relocalization of MTO fluorescence signal into the nucleus [52]. MTO nuclear localization is expected in necrotic cells. On the other hand, early apoptotic cells can retain mitochondrial membrane potential; however, the morphology of those mitochondria will be more granular instead of tubular shape [53]. Fluorescence images of MTO and AnnexinV/FITC in U87 MG cells in the presence of samples 2–4 display tubular mitochondria (in red), which are concentrated in clumps in the perinuclear area (see Figure 10, Figure 11 and Figure 12). Bright fluorescence of MTO was detected in mitochondria with the absence of MTO in the nuclei (counterstained with Hoechst—blue). This agrees with the MTT assay, which did not reveal the collapse of mitochondria and cell metabolic activity. Early apoptotic cells can be detected with the AnnexinV/FITC fluorescent probe, which is bound to phosphatidylserine expressed on the plasma membrane of cells [54,55,56,57]. Populations with and without AnnexinV/FITC labeling were observed in this study. A higher amount of phosphatidylserine foci (green) was observed in U87 MG cells in the presence of sample 3. Some spots were also detected at the sample 4 application. Phosphatidylserine localization in cells overlapped with the area of nanoparticle localization (see white arrows). In this approach, we cannot be completely sure of apoptosis as the main type of cell death. Many detailed studies are needed to be carried out in the future. However, this observation suggests that the most affected and probably damaged cells were cells with attached nanoparticle aggregates. In addition, with regard to the MTT assay, we assume that this nanoparticle-induced manifestation of cell death was ultimately defeated by cell defense processes and detoxication (drug, metals).

### 3.6. Naproxen Release

The release of the loaded naproxen from the samples MCM-41/Napro and MCM-41/Napro/Fe_2_O_3_ was studied in physiological intravenous saline solution (0.9% NaCl). The time dependence of naproxen release is shown in Figure 13. For both samples, the significant burst release was observed in the first 2 h, and two replicates were tested for each sample. We suppose that this release was driven by dissolution and was due to the release of the naproxen trapped on the external surface. When molecules of the solvent reached the surface, after wetting and non-bonding interactions between solvent molecules and the drugs, the drug molecules started to release from the surface. The released amount in 2 h represented 46% for the sample MCM-41/Napro and 34% for the sample MCM-41/Napro/Fe_2_O_3_. After 2 h, the slope of the curves changed. We suppose that after this time, the naproxen loaded in the pores was released, which slowed down the release. After 24 h, 79% of naproxen was released from the sample MCM-41/Napro and 68% from the sample MCM-41/Napro/Fe_2_O_3_. In the final stage of the release, between 24–72 h, the release continued, although a lower amount of naproxen was released in this time period. The total released amount after 72 h was 93% for the sample MCM-41/Napro and 79% for the sample MCM-41/Napro/Fe_2_O_3_. The release process from both samples was supposedly driven by wetting and the dissolution of the drug molecules present on the surface and in pores and diffusion of the solvents into the pores and diffusion of the dissolved drugs from the pores. A similar dependence of drug release due to diffusion/dissolution behavior in mesoporous silica microparticles was also shown in Salomen et al. [58] or Hu et al. [59]. The process of the release can be fitted by different kinetic models. Our results in previous works showed that, with the Fickian diffusion mechanism, kinetics mainly follows the diffusion mechanism dependent on Higuchi and Korsmeyer–Peppas kinetics models [23,60]. As it is obvious, the lower amount of the naproxen was released from hematite modified MCM-41. We suppose that this is due to the interaction of the drug with the surface of the metal oxide nanoparticles. Naproxen has an acidic character and thus its carboxylic group can interact with hematite nanoparticles. This interaction probably influences the lower total released amount from the sample MCM-41/Napro/Fe_2_O_3_, in comparison with the sample MCM-41/Napro.

## 4. Conclusions

In our study, we used MCM-41 silica with hexagonally ordered pores modified with iron oxide nanoparticles as a drug delivery system for drug naproxen. The magnetic properties of composite MCM-41/Napro/Fe_2_O_3_ are determined by the properties of Fe_2_O_3_ nanoparticles loaded within nanopores. After encapsulation of the magnetic nanoparticles into the mesoporous silica, the particles keep their superparamagnetic behavior and could be used for vectored drug delivery using magnetic fields and preparation of smart drug delivery systems. The superparamagnetic properties are caused by the small size of particles and the fact that each nanoparticle constitutes one single crystal. The sample MCM-41/Napro/Fe_2_O_3_ exhibits a high magnetic moment under the effect of a magnetic field, but no remanent magnetic moment is presented when the external magnetic field is removed. This property has an advantage in in vivo applications. After therapy using an external magnetic field, when the external magnetic field is switched off, the magnetic dipoles of the hematite nanoparticles randomize and zero net magnetic moment is spontaneously recovered. Moreover, the loading of the hematite into porous structure produces ultra-small superparamagnetic iron oxide nanoparticles (USPIONs), whose size is limited by the size of the silica cavities. Magnetic properties of sample diluted in saline solution (sample MCM-41/Napro/Fe_2_O_3_ in saline solution with pH = 7) confirmed that relaxation mechanism is caused predominantly by a Néel relaxation with a very small contribution of Brown relaxation. Realized magnetic study on studied sample after applying DC and AC magnetic fields with different frequencies suggested that encapsulation of nanoparticles into the porous system of mesoporous silica prevents the rotation of whole magnetic particle due to the “gripping effect”.

This is an interesting finding, which extends the advantages of using mesoporous support for magnetic nanoparticles from both a structural point of view and the point of regulating the relaxation process; this phenomenon has not been observed yet. Suppression of the Brown relaxation mechanism, which is undesirable in living systems for biomedical applications, may also be useful in magnetic hyperthermia.

The cytotoxicity tests were performed using human glioma U87 MG cells. The results demonstrate that the Fe_2_O_3_ nanoparticles did not induce cell death at the studied concentration if applied to cell culture media. However, our observations suggest that the most affected and probably damaged cells were labeled with AnnexinV/FITC and have nanoparticle aggregates attached to the cell surface. However, regarding MTT assays, the apoptotic stimulus triggered with oxidative stress was overcome by cell defense processes. This suggests that our studied nanocomposite system combines the advantage of mesoporous silica as a drug delivery system with magnetic nanoparticles.

## Figures and Tables

**Figure 1 nanomaterials-11-00901-f001:**
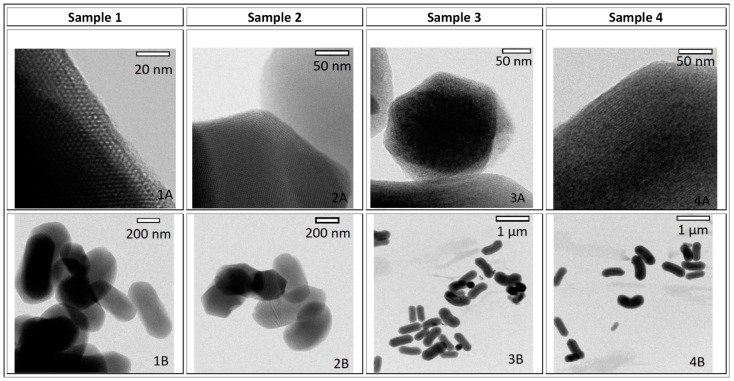
Sample 1—blank mesoporous silica MCM-41; sample 2—modified mesoporous silica MCM-41/Napro; sample 3—mesoporous silica containing iron oxide nanoparticles MCM-41/Fe_2_O_3_; sample 4—mesoporous silica with drug MCM-41/Napro/Fe_2_O_3_.

**Figure 2 nanomaterials-11-00901-f002:**
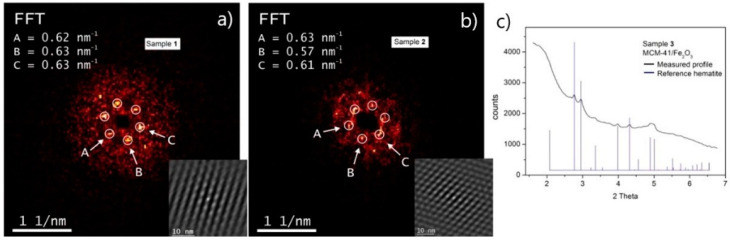
(**a**) Fast Fourier transform (FFT) Image associated with HRTEM of blank MCM-41 (sample 1 in Figure 1); pore axis aligned parallel with the electron beam. (**b**) FFT Image associated with HRTEM of the sample with drug MCM-41/Napro (sample 2 in Figure 1); pore axis aligned parallel with the electron beam/ (**c**) WAXS (Wide-angle X-ray scattering) spectrum of sample MCM-41/Fe_2_O_3_ (sample 3 in Figure 1), confirming the presence of Fe_2_O_3_ nanoparticles (hematite) in the composite. Measured sample profile (black line) was compared with the peaks of hematite reference, ICSD file No. 790007 (blue lines).

**Figure 3 nanomaterials-11-00901-f003:**
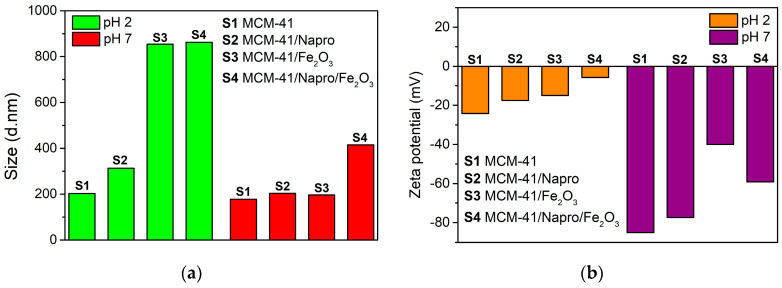
(**a**) Hydrodynamic size estimated from dynamic light scattering method of samples 1–4 at different pH: in simulated body fluid at pH = 7 (red) and simulated gastric fluid at pH = 2 (green). (**b**) Zeta potential (ZP) of samples 1–4 at different pH in simulated body fluid at pH = 7 (purple) and simulated gastric fluid at pH = 2 (orange).

**Figure 4 nanomaterials-11-00901-f004:**
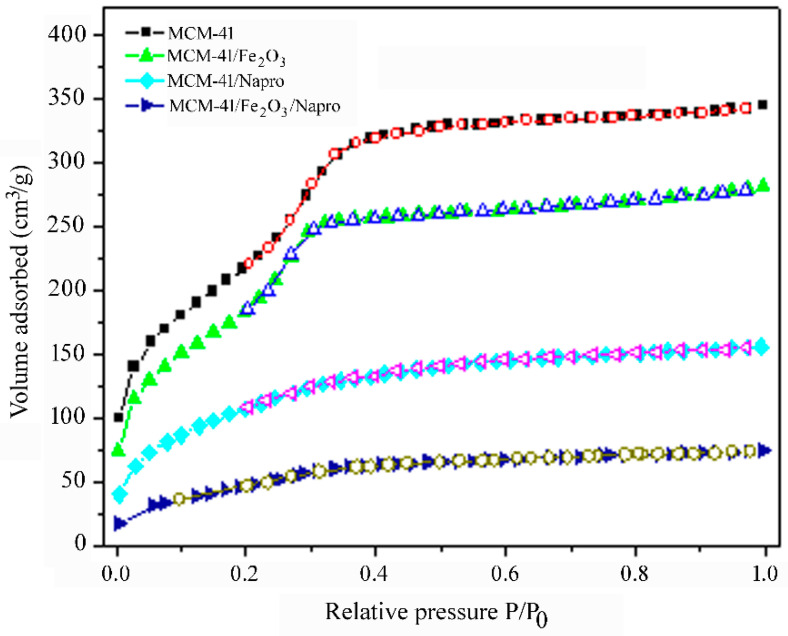
Black/red points show adsorption/desorption isotherms of the sample MCM-41 (sample 1), cyan/magenta points show adsorption/desorption isotherms of the sample MCM-41/Napro (sample 2), green/blue points show adsorption/desorption isotherms of the sample MCM-41/Fe_2_O_3_ (sample 3), and royal/grey points show adsorption/desorption isotherms of the sample MCM-41/Napro/Fe_2_O_3_ (sample 4).

**Figure 5 nanomaterials-11-00901-f005:**
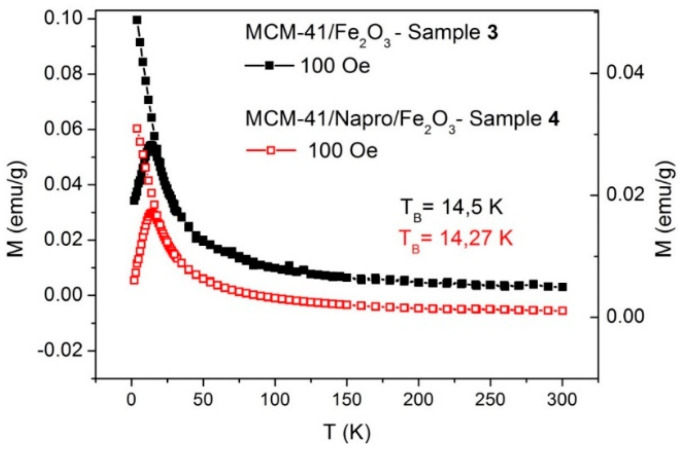
Magnetization vs. temperature of sample 3 and sample 4 at the external magnetic field of 100 Oe, measured in zero-field-cooling (ZFC) and field cooling (FC) regimes.

**Figure 6 nanomaterials-11-00901-f006:**
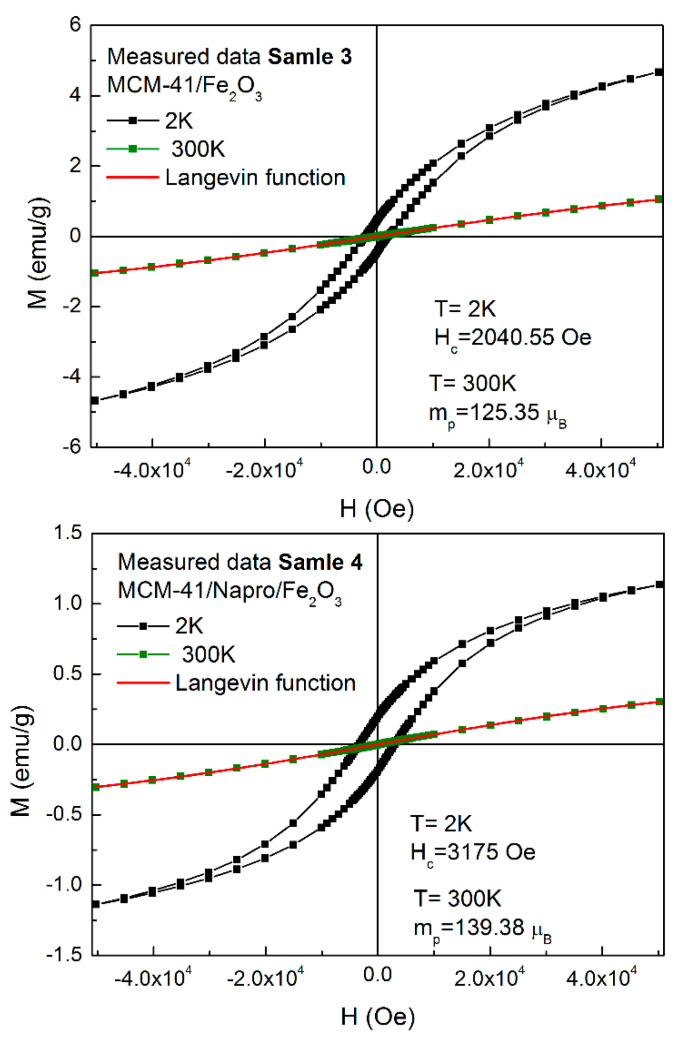
Magnetization curves (M vs. H) of samples 3 and 4 at different temperatures (2 K-black squares, 300 K-green squares) below and above the blocking temperature T_B_. The red line represents the Langevin fit of experimental data in a superparamagnetic state (above blocking temperature).

**Figure 7 nanomaterials-11-00901-f007:**
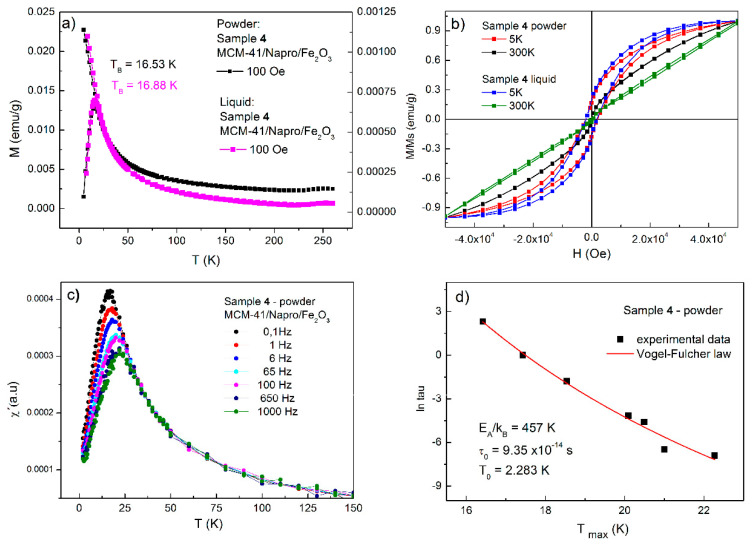
Comparison of magnetic properties of powdered and by liquid diluted sample 4. (**a**) ZFC/FC magnetization for powdered (black squares) and for liquid diluted (magenta squares) sample 4. (**b**) Magnetization curves (M vs. H) of sample 4 at different temperatures 5 K and 300 K for powdered and liquid diluted form. (**c**) Real par of AC susceptibilty vs. temperature at different frequencies. (**d**) Calculated experimental data using Vogel-Fulcher law.

**Figure 8 nanomaterials-11-00901-f008:**
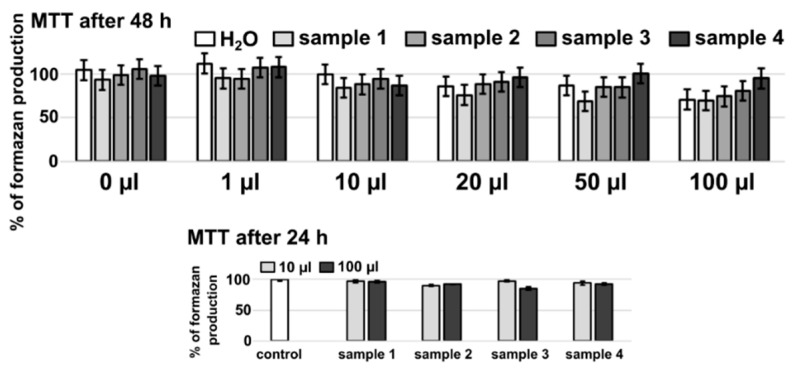
Production of formazan (cell metabolic activity related to the viability of cells) in U87 MG cells in the absence and presence of (0–100) µL/mL of distilled water with samples 1–4. Substances were incubated with cells for 24 and 48 h.

**Figure 9 nanomaterials-11-00901-f009:**
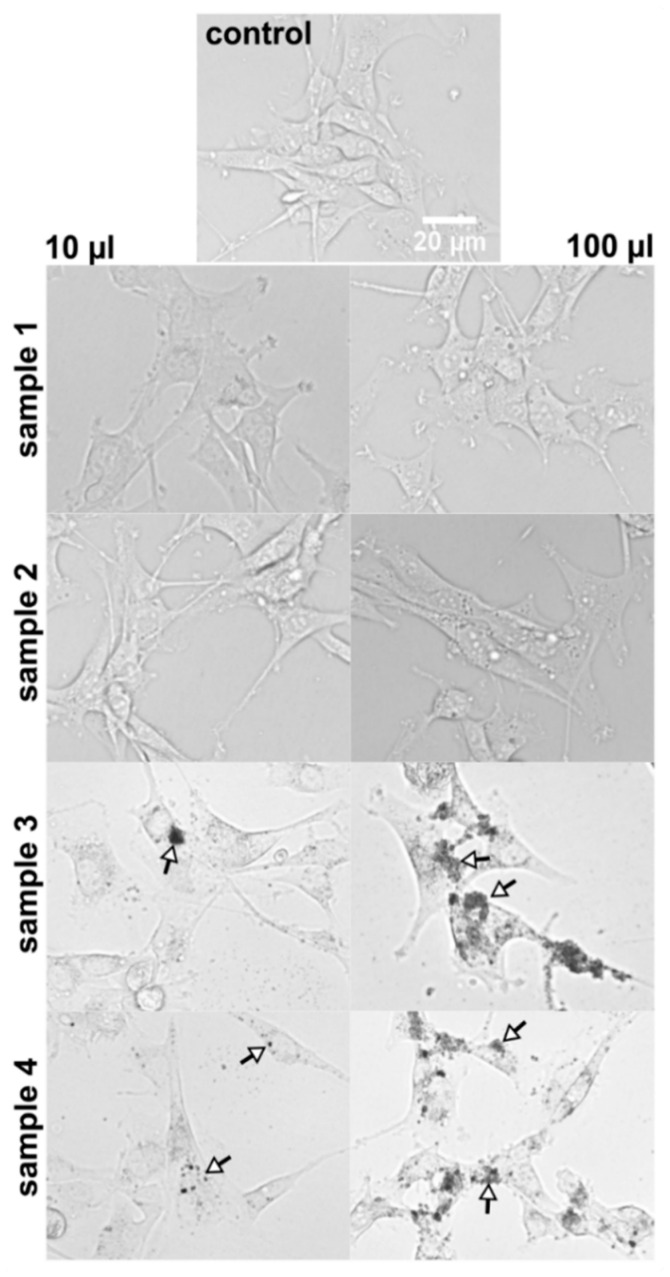
Bright-field images of U87 MG cells in the absence and presence of samples 1–4 at concentrations 10 and 100 µL/mL. Images were detected 24 h after substance administration. Arrows point to nanoparticle aggregates at 10 µL/mL concentrations. The clusters of nanoparticles were adsorbed to the cell surface at 100 µL/mL.

**Figure 10 nanomaterials-11-00901-f010:**
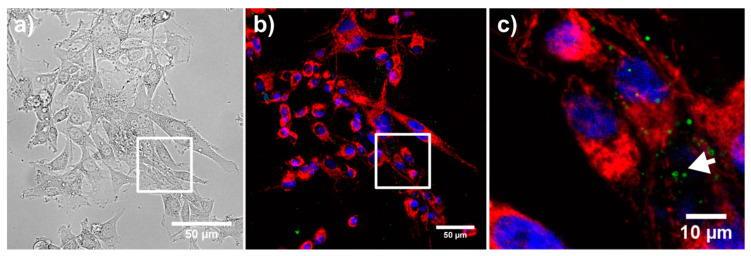
Bright-field (**a**) and fluorescence images (**b**,**c**) of U87 MG cells in the presence of 100 µL/mL of sample 2. White square selection of the fluorescence image (**b**) was zoomed in (**c**). The white arrow points to externalized phosphatidylserine found in few cells. Red fluorescence corresponds to mitochondrial probe MitoTracker Orange CMTM/Ros (MTO, 15 min, 200 nM), green fluorescence represent phosphatidylserine labeled with AnnexinV/FITC probe (15 min, 5 µL/mL), and blue fluorescence corresponds to nuclei labeled with Hoechst (15 min, 10 µg/mL).

**Figure 11 nanomaterials-11-00901-f011:**
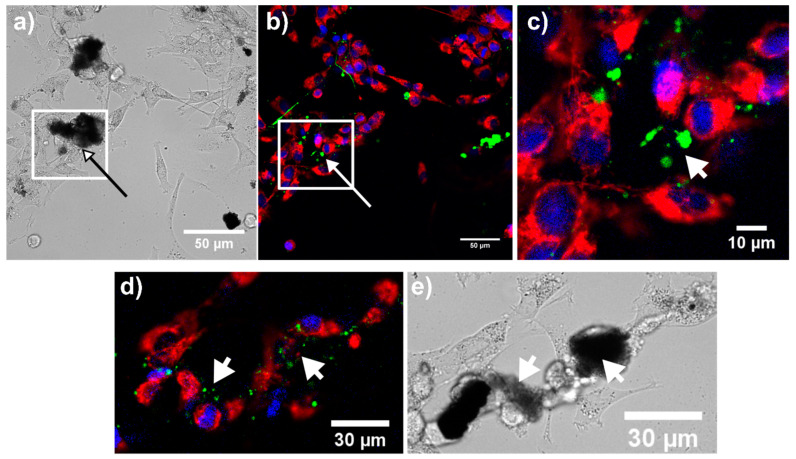
Bright-field (**a**,**e**) and MTO (red), AnnexinV/FITC (green), and Hoechst (blue) fluorescence (**b**–**d**) images of U87 MG cells in the presence of 100 µL/mL of sample 3. White square selection of the fluorescence image (**b**) was zoomed in (**c**). The white arrow points to externalized phosphatidylserine found at the same place as the aggregates of nanoparticles. Externalization of phosphatidylserine in cells refers to early apoptosis induction.

**Figure 12 nanomaterials-11-00901-f012:**
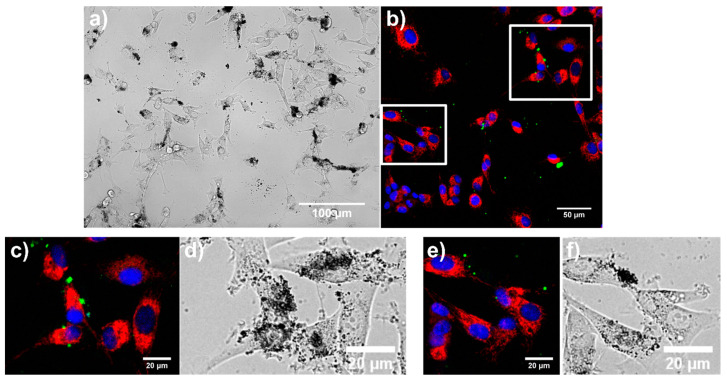
Bright-field (**a**,**d**,**f**) and MTO (red), AnnexinV/FITC (green), and Hoechst (blue) fluorescence (**b**,**c**,**e**) images of U87 MG cells in the presence of 100 µL/mL of sample 4. White square selections of the fluorescence image (**b**) were zoomed in (**c**,**e**). Few spots of externalized phosphatidylserine could be observed in the zoomed images.

**Figure 13 nanomaterials-11-00901-f013:**
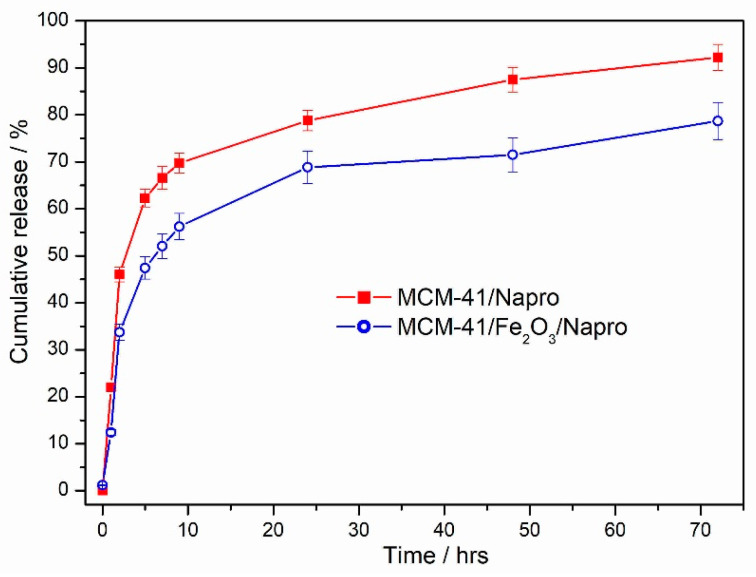
The time dependence of naproxen release from samples 3 and 4.

**Table 1 nanomaterials-11-00901-t001:** Textural characteristics of the studied materials.

Sample	BET Surface Area (m^2^/g)	External Surface Area (m^2^/g)	Pore Volume (cm^3^/g)	Pore Diameter (Å)
1—MCM-41	820	62	0.642	35
2—MCM-41/Napro	295	83	0.134	25
3—MCM-41/Fe_2_O_3_	670	51	0.516	32
4—MCM-41/Napro/Fe_2_O_3_	159	74	0.067	-

## Data Availability

Samples of the compounds are available from the authors.

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
