# Peer review of "Magnetic Characterization and Moderate Cytotoxicity of Magnetic Mesoporous Silica Nanocomposite for Drug Delivery of Naproxen"

_nanomaterials, 2021, doi:10.3390/nano11040901_

Round 1

Reviewer 1 Report

The manuscript nanomaterials-1107495  "Relation between Cytotoxicity and Magnetic Characterization of Magnetic Mesoporous Silica Nanocomposite for Drug Delivery of Naproxen" by Zelenakova and co-workers describes the creation of drug delivery composite system of mesoporous silica containing the iron oxide nanoparticles and naproxen drug. The resulting nanomaterials were characterized by TEM, DLS, FT-IR, nitrogen adsorption/desorption, and their cytotoxicity on human glioma U87 MG cells was studied. This work is a successful continuation of the previous research of the authors, despite the fact that the authors have already shown the results of this work at conferences several years ago.

Questions and comments:

1) The manuscript is titled "Relation between Cytotoxicity and Magnetic Characterization ...", but the paper does not show a direct relationship between toxicity and magnetic properties. Authors need to change the title or show this relationship.

2) I propose to transfer part of the data (for example, FTIR spectra in Figure 6) to supplementary materials due to the large size of the manuscript.

3) A large number of self-citation (12/50 references) in the manuscript should be corrected.

4) English must be checked again by a native speaker.

Author Response

Responses to the Reviewer 1 and list of the revisions applied in the manuscript 1107495 “Magnetic Characterization and Cytotoxicity of Magnetic Mesoporous Silica Nanocomposite for Drug Delivery of Naproxen”

 Review 1

Comment 1: The manuscript is titled "Relation between Cytotoxicity and Magnetic Characterization ...", but the paper does not show a direct relationship between toxicity and magnetic properties. Authors need to change the title or show this relationship.

Answer 1: We thank the reviewer for this comment. The title of the article was changed to better express the content.

Comment 2: I propose to transfer part of the data (for example, FTIR spectra in Figure 6) to supplementary materials due to the large size of the manuscript.

Answer 2: Figure of the infrared spectra was moved to the Supplementary information (Appendix A).

Comment 3:  A large number of self-citation (12/50 references) in the manuscript should be corrected.

Answer 3: We are sorry for the use of the redundant works. We rationalized a number of self-citations, 4 of them were deleted from the references.

Comment 4: English must be checked again by a native speaker.

Answer 4: The manuscript was re-checked, the language corrections were made.

Reviewer 2 Report

The manuscript by Zelenakova et al presents the preparation and characterisation of silica-based magnetic nanocomposites as a drug delivery system. From the conceptual point of view, no justification with experimental proof of the need of magnetic materials in this system is presented. In the intro and conclusion authors mention magnetic guidance and potentially magnetic hyperthermia, but no tests are presented at this respect. Also hematite would not be a suitable material for these application due to its inadequate magnetic properties.

More concrete concerns follow:

  • TEM images shown are either not good enough or the system does not have the properties reported. The presence of Fe2O3 can not be clearly seen and it cannot be unequivocally placed inside the pores as the authors claim. In samples 2, 3 and 4 no pores are visible.
  • 'From these observations it is obvious, that silica matrix is responsible for controlled growth of Fe2O3 MNPs and served as nanoreactor for preparation of superparamagnetic iron oxide nanoparticles (SPIONs) with very low particle size distribution'. This observation is not obvious at all from TEM images presented. No size distribution of Fe2O3 particles is provided (from the images the presence of these particles is not clear at all)
  • Why was the FFT study only carried out in samples 1 and 2? why not in 3 and 4 also?
  • WAXS spectra might prove the presence of Fe2O3 but it doesn't say anything about its location (inside pores, outside pores, independent from the SiO2 structure...)
  • 'This outcome suggests that no agglomeration of MCM-41 particles took place in acidic or basic medium'. No basic pH was tested
  • Iron oxide is not stable at low pH values. How does this impact DLS/Zpot observations?
  • They observe agglomeration at low pH. Why?
  • Explain where the negative charge in Zpot comes from. If Napro and hematite nanoparticles are inside the SiO2 structure, why are all samples different in surface charge if the outer structure according to the authors is the same?
  • In the FTIR, why are Fe-O bands not visible at all, while all Napro bands are visible?
  • In section 3.5.2., with the images acquired/provided it is not possible to judge the location of the particles (inside, attached to the membrane...). Confocal Zstack series would be needed for that purpose.
  • 'Interestingly, round vesicular objects were detected in within the cell’s cytoplasm at presence of 100 μl/ml of S2.' Unfortunately these vesicular objects cannot be unequivocally associated with the particles.
  • section 3.5.3 is full of hypothesis without a single proof to support it. All of the claims they make can and should be proved with further tests.
  • In this section they talk about clear apoptotic signs, but they didn´t see any effect on the cytotoxicity tests
  • In section 3.6 they again hypothesise about mechanisms. There are mathematical models of release that can be used to fit the data and extract conclusions about the release mechanisms.
  • In the conclusions the authors speak about the high magnetic moment of their composites and about the suitability of hematite for biomed applications. Hematite is the least desirable stable form of iron oxide for this kind of applications as mentioned in many references.

Author Response

Responses to the Reviewer 2 and list of the revisions applied in the manuscript 1107495 “Magnetic Characterization and Cytotoxicity of Magnetic Mesoporous Silica Nanocomposite for Drug Delivery of Naproxen”

Review 2

Comment 1: TEM images shown are either not good enough or the system does not have the properties reported. The presence of Fe2O3 can not be clearly seen and it cannot be unequivocally placed inside the pores as the authors claim. In samples 2, 3 and 4 no pores are visible.

Answer 1: We can not agree with this statement. On Fig. 2 (1A) the periodic porous structure with alternation of pores and pore walls is clearly visible. The same in Fig. 2 (2A), where the presence it is obvious. Although in the latter the lower magnification was used, still the pores can be distinguished. The presence of magnetic nanoparticles in the pores of silica matrix is in Fig 2 (3A, 4A) clearly seen as a dark spots. This is typical observation of nanoparticles incorporated in porous matrix.

Comment 2: 'From these observations it is obvious, that silica matrix is responsible for controlled growth of Fe2O3 MNPs and served as nanoreactor for preparation of superparamagnetic iron oxide nanoparticles (SPIONs) with very low particle size distribution'. This observation is not obvious at all from TEM images presented. No size distribution of Fe2O3 particles is provided (from the images the presence of these particles is not clear at all)

Answer 2: The observation is clearly visible from HRTEM images, WAXS measurements as well as magnetic measurements. See also the answer to Comment 1.

Comment 3: Why was the FFT study only carried out in samples 1 and 2? why not in 3 and 4 also?

Answer 3: For sample with magnetic nanoparticles WAXS method was used, which was straightforward for identification of iron oxide phase.

Comment 4: WAXS spectra might prove the presence of Fe2O3 but it doesn't say anything about its location (inside pores, outside pores, independent from the SiO2 structure...)

Answer 4: The features observed in presented WAXS spectra are typical for nanoparticles embedded in porous silica matrix (broadening of the diffraction peaks, their overlapping with signal of amorphous silica). For comparison see also:

  1. Zeleňák et al.: RSC Advances, 9 (2019) 3679-3687.
  2. Zeleňáková et al.: Scientific Reports, 9 (2019) 15852.

Comment 5: 'This outcome suggests that no agglomeration of MCM-41 particles took place in acidic or basic medium'. No basic pH was tested.

Answer 5: Yes, we are sorry. The pH was 7, we corrected the text and the word “basic” was changed to “physiological pH”.

Comment 6: Iron oxide is not stable at low pH values. How does this impact DLS/Zpot observations?

Answer 6: No significant hematite leaching was observed during the DSL experiment. It is well known, that in acidic medium hematite is more stable than e.g. magnetite or ferrites. Hematite is more stable in inorganic acids (like HCl), than in organic acids. The nanoparticles of hematite were prepared by thermal decomposition of iron(III) nitrate at high temperatures so a number of reactive surface hydroxyls was reduced. The DSL experiments were done at room temperature, in diluted acid with the concentration 0,01M (pH = 2) and the measurements took place for several minutes only. No leaching was observed. The samples colour and colour of the solution after measurements remained unchanged (brown-orange and colourless, respectively). 

Comment 7: They observe agglomeration at low pH. Why?

Answer 7: At low pH, the protonation of surface hydroxyls of MCM-41 silica and presence of Cl- anions (HCl was used to adjust the pH to 2) may lead to the non-covalent interactions between silica nanoparticles, which enhances the agglomeration. The information was added into the manuscript.

Comment 8: Explain where the negative charge in Zpot comes from. If Napro and hematite nanoparticles are inside the SiO2 structure, why are all samples different in surface charge if the outer structure according to the authors is the same?

Answer 8: Discussion, page 7, in the manuscript was revised as follows. The isoelectric points of hematite and silica are different (pH ~ 7.8 for hematite and pH ~2 for ideal bulk silica) but may be a little shifted for different samples. Therefore, the zeta potential (ZP) is an interplay of the surface charges of both these systems. The silica matrix phase is in abundance. At pH = 7 silica particles are negatively charged while hematite positively. As a result, silica samples 1 and 2, containing no iron nanoparticles have larger negative ZP than samples 3 and 4, containing hematite. The same holds for pH = 2. Since we are closer to isoelectric point of silica (around pH = 2 for the bulk silica), the ZP significantly increases in comparison with pH = 7. Samples containing hematite are closer to zero ZP value characteristic for isoelectric point.

Comment 9: In the FTIR, why are Fe-O bands not visible at all, while all Napro bands are visible?

Answer 9: It is to note, that hematite nanoparticles are incorporated in the channels of amorphous silica matrix, which dominates, and stretching vibration of the hematite can not be distinguished in the spectra. Moreover, the KBr technique, used for measurements, also diminishes this identification. The discussion was added to the manuscript.

Comment 10: In section 3.5.2., with the images acquired/provided it is not possible to judge the location of the particles (inside, attached to the membrane...). Confocal Zstack series would be needed for that purpose.

Answer 10: We apologize for a mistake in the interpretation. Due to the size of the nanoparticles, we assume that most of the clouds of these nanoparticles are adsorbed to cell surface. This effect was observed from the 3D z-stacks of bright field images.

Comment 11: 'Interestingly, round vesicular objects were detected in within the cell’s cytoplasm at presence of 100 μl/ml of S2.' Unfortunately these vesicular objects cannot be unequivocally associated with the particles.

Answer 11: We hypothesize that these vesicular objects could be associated with autophagic vesicles and process of cell detoxication. It is difficult to distinguish the presence of nanoparticles inside the vesicles. However, they can be form because of naproxen interaction with cells.

Comment 12: section 3.5.3 is full of hypothesis without a single proof to support it. All of the claims they make can and should be proved with further tests.

Answer 12: This section was reformulated to be more obvious. The reformulated text is as follows:

“High level of nanoparticles biocompatibility with glioma cells was demonstrated above by means of cell morphology and metabolic activity. However, vesicles formation in cells suggests certain degree of intoxication caused by presence of the nanoparticles. MTT-assay is closely connected with mitochondria where formazan is produced [48]. For this reason, mitochondria of U87 MG cells were labeled with mitochondrial probe MTO. MTO is partially sensitive to mitochondrial membrane potential. Dissipation of mito-chondrial membrane potential induced by toxins can be observed in cells by decreasing of MTO fluorescence intensity up to relocalization of MTO fluorescence signal into the nu-cleus [49]. MTO nuclear localization is expected in necrotic cells. On the other hand, early apoptotic cells can retain mitochondrial membrane potential, however, morphology of those mitochondria will be more granular instead of tubular shape [50]. Fluorescence im-ages of MTO and AnnexinV/FITC in U87 MG cells in the presence of samples 2-4 dis-played tubular mitochondria (in red), which are concentrated in clumps in the perinuclear area, see Figures 11-13. Bright fluorescence of MTO was detected in mitochondria with the absence of MTO in the nuclei (counterstained with Hoechst - blue). This agrees with MTT-assay that did not reveal collapse of mitochondria and cell metabolic activity. Early apoptotic cells can be detected with AnnexinV/FITC fluorescent probe, which is bound to phosphatidylserine expressed on plasma membrane of cells [51]. Populations with and without AnnexinV/FITC labeling were observed in this study. The higher amount of phosphatidylserine foci (green) was observed in U87 MG cells in the presence of sample 3. Some spots were also detected at sample 4 application. Phosphatidylserine localization in cells overlapped with the area of nanoparticles localization (see white arrows). This ob-servation suggests that the most affected cells by early apoptosis were those with attached aggregates of nanoparticles. However, regarding to MTT-assay, we assume that the apop-totic stimulus triggered in cells was finally defeated by cell defense processes, and detoxi-cation (drug, metals).”

  1. Mosmann, T. Rapid colorimetric assay for cellular growth and survival: application to proliferation and cytotoxicity assays. J. Immunol. Methods. 1983 vol. 65, 55-63.
  2. Scorrano, L.; Petronilli, V.; Colonna, R.; Di Lisa, F.; Bernardi, P. Chloromethyltetramethylrosamine (Mitotracker Orange) induces the mitochondrial permeability transition and inhibits respiratory complex I. Implications for the mechanism of cytochrome c release. J. Biol. Chem. 1999, vol. 274, 24657-63.
  3. Tomkova, S.; Misuth, M.; Lenkavska, L.; Miskovsky, P.; Huntosova, V. In vitro identification of mitochondrial oxidative stress production by time-resolved fluorescence imaging of glioma cells. Biochimica et Biophysica Acta (BBA) - Molecular Cell Research 2018, vol. 1865, Pages 616-628.
  4. Brauchle, E.; Thude, S.; Brucker, S.Y.; Schenke-Layland, K. Cell death stages in single apoptotic and necrotic cells monitored by Raman microspectroscopy. Scientific Reports 2014, vol. 4, 4698.

Comment 13: In this section they talk about clear apoptotic signs, but they didn´t see any effect on the cytotoxicity tests

Answer 13: In the present study, we have applied AnnexinV-FITC probe to labelled phosphatidylserine expressed on plasma membrane of early apoptotic cells. We can see that the morphology of cells is still compact. Moreover, mitochondrial sensor refers that mitochondria of cells keep mitochondrial membrane potential. As MTT-assay is based on formazan production in the live mitochondria and reflects metabolic activity of cells, the observed results agree with MTT-assay results. This section was reformulated to be more obvious.

Comment 14: In section 3.6 they again hypothesise about mechanisms. There are mathematical models of release that can be used to fit the data and extract conclusions about the release mechanisms.

Answer 14: The process of the release can be fitted by different kinetic models. Our results in previous works showed, that mainly follows diffusion Higuchi and Korsmeyer-Peppas dependent kinetics models, with Fickian diffusion mechanism. Such short discussion, with references was added into the manuscript.

Also, the Fig. 14 was actualised to show the standard deviations. Two replicates were tested.

Figure 14. The time dependence of naproxen release from the samples 3 and 4.

Comment 15: In the conclusions the authors speak about the high magnetic moment of their composites and about the suitability of hematite for biomed applications. Hematite is the least desirable stable form of iron oxide for this kind of applications as mentioned in many references.

Answer 15: Hematite is the most stable and extensively studied iron oxide. Bulk hematite is crystal with a corundum structure containing two sublattices creating an antiferromagnetic system. The magnetic moment of bulk hematite and microsized particles is very small and therefore unsuitable for biomedical applications. 

With decreasing size of hematite particles below to 10 nm, the magnetic moment of particles drastically increases. They show slight canting considering the basal plane, resulting to a magnetic moment, which originates from the superexchange interaction. Also was observed the absence of MORIN temperature TM for the particles with size 8-20 nm. Due to the suppression of the Morin transition in small hematite nanoparticles (below 20 nm), only a high temperature phase can exist, in which the net magnetic moment originates in small spin canting away from antiferromagnetic alignment.  Thus, regard to the above, the magnetic moment of nanosized hematite is comparable to other forms of iron(III) oxide.

This  Discussion in the manuscript was added.

Reviewer 3 Report

The manuscript describes the use of MCM-41 silica with magnetic inclusions in the form of the ultra-small magnetic nanoparticles of α-Fe2O3 acting as a carrier for drug delivery. At the same time, the interaction of such material with cells is discussed concerning the performed in-vitro cytotoxicity tests. I can only confirm that this is a very valuable study allowing further work on target drug delivery. Data from apoptosis measurements that are presented are very interesting and promising for applications. They can be also important for magnetic hyperthermia applications. The results of the presented research are based on a wide range of research methods, which in addition to valuable results makes this work important as up to an extent, they can be compared to results generated by other research groups. In particular, the authors should note that some preliminary results on the study of the magnetic properties of silica with ultra-small magnetic nanoparticles based on iron oxides in silica pores were already published with an example being the work by B. Zapotoczny et al. in Journal of Magnetism and Magnetic Materials 374 (2015) 96–102, where the fluorescein was used to show the drug release efficiency.

I recommend this manuscript for publication. I suggest only minor additions to the literature references to publications concerning the use of mesoporous silica with ultra-small magnetic nanoparticles.

Author Response

Responses to the Reviewer 3 and list of the revisions applied in the manuscript 1107495 “Magnetic Characterization and Cytotoxicity of Magnetic Mesoporous Silica Nanocomposite for Drug Delivery of Naproxen”

 Review 3:

Comment 1: The manuscript describes the use of MCM-41 silica with magnetic inclusions in the form of the ultra-small magnetic nanoparticles of α-Fe2O3 acting as a carrier for drug delivery. At the same time, the interaction of such material with cells is discussed concerning the performed in-vitro cytotoxicity tests. I can only confirm that this is a very valuable study allowing further work on target drug delivery. Data from apoptosis measurements that are presented are very interesting and promising for applications. They can be also important for magnetic hyperthermia applications. The results of the presented research are based on a wide range of research methods, which in addition to valuable results makes this work important as up to an extent, they can be compared to results generated by other research groups. In particular, the authors should note that some preliminary results on the study of the magnetic properties of silica with ultra-small magnetic nanoparticles based on iron oxides in silica pores were already published with an example being the work by B. Zapotoczny et al. in Journal of Magnetism and Magnetic Materials 374 (2015) 96–102, where the fluorescein was used to show the drug release efficiency.

Answer 1: We thank the reviewer for the comment. The respective work was cited in the manuscript as the reference 39.

Comment 2: I recommend this manuscript for publication. I suggest only minor additions to the literature references to publications concerning the use of mesoporous silica with ultra-small magnetic nanoparticles.

Answer 2: We added the reference mentioned in the previous comment.

Round 2

Reviewer 2 Report

I feel that the authors have made an effort to improve the quality of the manuscript. However, the issues regarding the structural characterisation of the nanomaterials persist (black poorly defined dots do not necessarily correspond to iron oxide particles, there is no analysis on the size of this particles from TEM...). There are also some inconsistencies in other data (iron oxide NP contribute to Zpot but they do not contribute to FTIR according to the authors). Also, their explanation on the properties of nano-hematite can be correct and enlightening, but hematite is still the least desirable form of stable iron oxide for biomed applications because its magnetic properties are poorer than those of the other forms (magnetite, maghemite).

Sadly, these points together with the lack of novelty of the study, makes me recommend that the manuscript be rejected.

Author Response

Responses to the Reviewer 2 and list of the revisions applied in the manuscript 1107495 “Magnetic Characterization and Cytotoxicity of Magnetic Mesoporous Silica Nanocomposite for Drug Delivery of Naproxen”- 2 round

Review 2

Comment 1: „...inconsistencies in other data (iron oxide NP contribute to Zpot but they do not contribute to FTIR according to the authors).“ 

Answer 1A: There is difference between the methods. The Zpot was measured in the solution. Therefore, the fluid can diffuse into the pores, can diffuse also to the iron oxide NPs and therefore the Zpot reflects the surface charges of both silica and hematite NPs. The isoelectric points of hematite and silica are different (pH ~ 7.8 for hematite and pH ~2 for ideal bulk silica) but may be a little shifted for different samples. Therefore, the zeta potential (ZP) is an interplay of the surface charges of both these systems. The silica matrix phase is in abundance. At pH = 7 silica particles are negatively charged while hematite positively. As a result, silica samples 1 and 2, containing no iron nanoparticles have larger negative ZP than samples 3 and 4, containing hematite. The same holds for pH = 2. Since we are closer to isoelectric point of silica (around pH = 2 for the bulk silica), the ZP significantly increases in comparison with pH = 7. Samples containing hematite are closer to zero ZP value characteristic for isoelectric point.

Answer 1B: As for FTIR spectra, the situation is different. They are measured in solid state. The hematite NPs are hidden by silica matrix. The IR beam is first absorbed by the silica walls on the surface. Therefore, the IR response of hematite nanoparticles embedded in silica is too weak to be observed. The absorption bands in the IR spectra in the region about 500 cm-1 are stronger for silica, than hematite. Therefore, stretching vibration Fe-O of the hematite NPs cannot be distinguished in the spectra.

Discussion in manuscript was appended about mentioned information.

Comment 2: the comment to TEM

Answer 2: We cannot agree with the statement of reviewer. On Fig. 2 (A) the periodic porous structure with alternation of pores and pore walls is clearly visible. The presence of magnetic nanoparticles in the pores of silica matrix is in Fig 2 (3A, 4A) clearly seen as a dark spots. For the better observation we have changed the picture 3A in Fig. 2. The new Fig.2 was appended in manuscript, see below.

 New Figure2.

 The confirmation of presence of magnetic nanoparticles is by the magnetic measurements there are very sensitive on the structural characteristics. The value of magnetic moment of magnetic nanoporous composite is comparable with the magnetic moment of magnetic assembles, composed from isolated magnetic particles with the same size and structural characteristics.

 Regarding with the size of iron oxide nanoparticles, it is evident, that the size of particles is determined by the size of porous system, since the growth of magnetic nanoparticles is controlled by porous silica matrix.
